# CAT3D: Create Anything in 3D with Multi-View Diffusion Models

**Ruiqi Gao**[1]* **Aleksander Hołyński**[1]* **Philipp Henzler**[2] **Arthur Brussee**[1]
**Ricardo Martin-Brualla**[2] **Pratul Srinivasan**[1] **Jonathan T. Barron**[1] **Ben Poole**[1]*
[1]Google DeepMind [2]Google Research *equal contribution

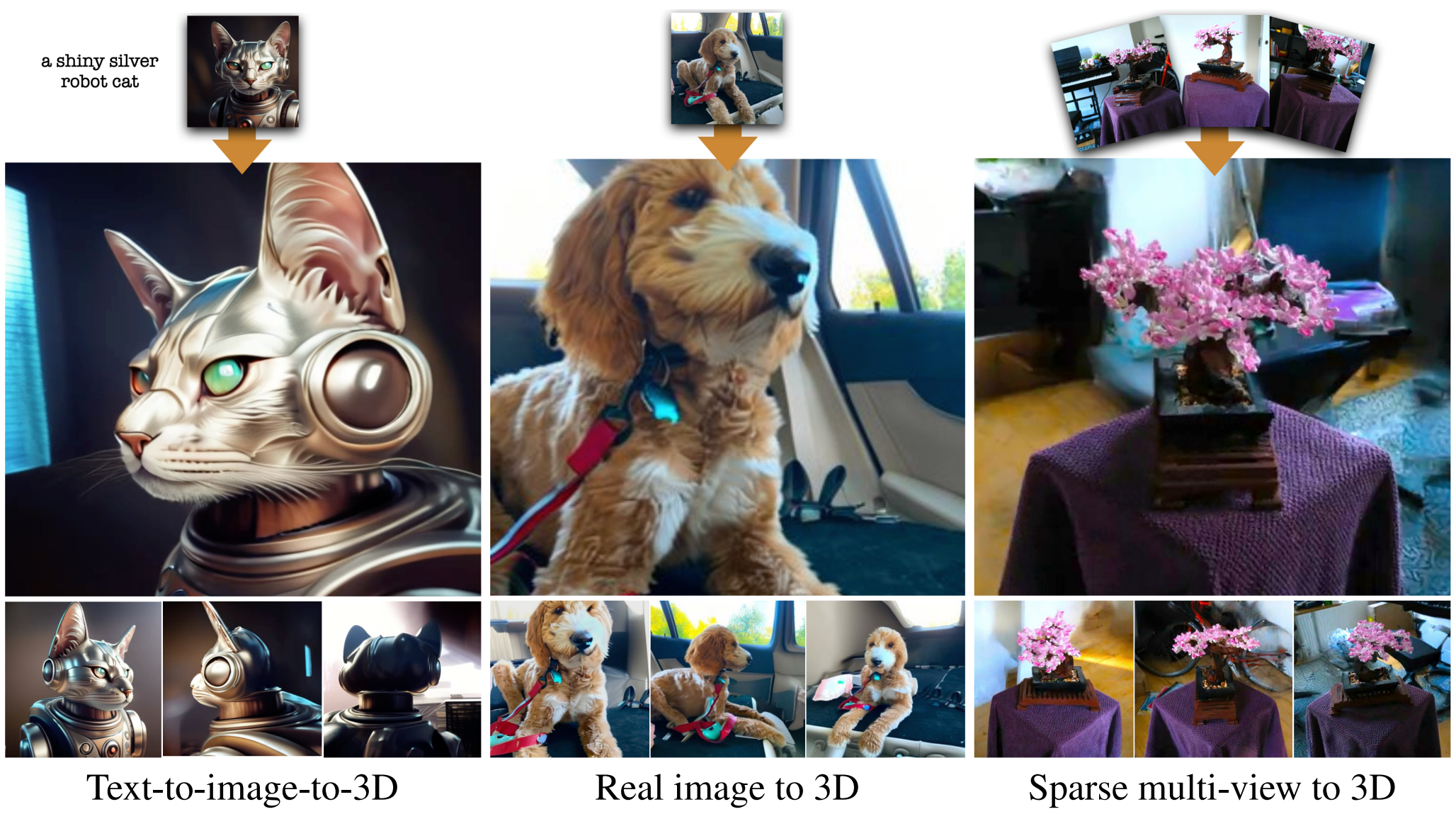

Figure 1: CAT3D enables 3D scene creation from any number of generated or real images.

## Abstract

Advances in 3D reconstruction have enabled high-quality 3D capture, but require a user to collect hundreds to thousands of images to create a 3D scene. We present CAT3D, a method for creating anything in 3D by simulating this real-world capture process with a multi-view diffusion model. Given any number of input images and a set of target novel viewpoints, our model generates highly consistent novel views of a scene. These generated views can be used as input to robust 3D reconstruction techniques to produce 3D representations that can be rendered from any viewpoint in real-time. CAT3D can create entire 3D scenes in as little as one minute, and outperforms existing methods for single image and few-view 3D scene creation.

## 1 Introduction

The demand for 3D content is higher than ever, since it is essential for enabling real-time interactivity for games, visual effects, and wearable mixed reality devices. Despite the high demand, high-quality 3D content remains relatively scarce. Unlike 2D images and videos which can be easily captured with consumer photography devices, creating 3D content requires complex specialized tools and a substantial investment of time and effort.

38th Conference on Neural Information Processing Systems (NeurIPS 2024).

Fortunately, recent advancements in photogrammetry techniques have greatly improved the accessibility of 3D asset creation from 2D images. Methods such as NeRF [1], Instant-NGP [2], and Gaussian Splatting [3] allow anyone to create 3D content by taking photos of a real scene and optimizing a representation of that scene's underlying 3D geometry and appearance. The resulting 3D representation can then be rendered from any viewpoint, similar to traditional 3D assets. Unfortunately, creating detailed scenes still requires a labor-intensive process of capturing hundreds to thousands of photos. Captures with insufficient coverage of the scene can lead to an ill-posed optimization problem, which often results in incorrect geometry and appearance and, consequently, implausible imagery when rendering the recovered 3D model from novel viewpoints.

Reducing this requirement from dense multi-view captures to less exhaustive inputs, such as a single image or text, would enable more accessible 3D content creation. Prior work has developed specialized solutions for different input settings, such as geometry regularization techniques targeted for sparse-view reconstruction [4, 5], feed-forward models trained to create 3D objects from single images [6], or the use of image-conditioned [7] or text-conditioned [8] generative priors in the optimization process—but each of these specialized methods comes with associated limitations in quality, efficiency, and generality.

In this paper, we instead focus on the fundamental problem that limits the use of established 3D reconstruction methods in the observation-limited setting: an insufficient number of supervising views. Rather than devising specialized solutions for different input regimes [8, 9, 7], a shared solution is to instead simply *create* more observations—collapsing the less-constrained, under-determined 3D creation problems to the fully-constrained, fully-observed 3D reconstruction setting. This way, we reformulate a difficult ill-posed *reconstruction* problem as a *generation* problem: given any number of input images, generate a collection of consistent novel observations of the 3D scene. Recent video generative models show promise in addressing this challenge, as they demonstrate the capability to synthesize video clips featuring plausible 3D structure [10, 11, 12, 13, 14, 15]. However, these models are often expensive to sample from, challenging to control, and limited to smooth and short camera trajectories.

Our system, **CAT3D**, instead accomplishes this through a multi-view diffusion model trained *specifically* for novel-view synthesis. Given any number of input views and any specified novel viewpoints, our model generates multiple 3D-consistent images through an efficient parallel sampling strategy. These generated images are subsequently fed through a robust 3D reconstruction pipeline to produce a 3D representation that can be rendered at interactive rates from any viewpoint. We show that our model is capable of producing photorealistic results of arbitrary objects or scenes from any number of captured or synthesized input views in as little as one minute. We evaluate our work across various input settings, ranging from sparse multi-view captures to a single captured image, and even just a text prompt (by using a text-to-image model to generate an input image from that prompt). CAT3D outperforms prior works for measurable tasks (such as the multi-view capture case) on multiple benchmarks, and is an order of magnitude faster than previous state-of-the-art. For tasks where empirical performance is difficult to measure (such as text-to-3D and single image to 3D), CAT3D compares favorably with prior work in all settings.

## 2 Related Work

Creating entire 3D scenes from limited observations requires 3D generation, *e.g.*, creating content in unseen regions, and our work builds on the ever-growing research area of 3D generative models [16]. Due to the relative scarcity of 3D datasets, much research in 3D generation is centered on transferring knowledge learned by 2D image-space priors, as 2D data is abundant. Our diffusion model is built on the recent development of video and multi-view diffusion models that produce highly consistent novel views. We show that pairing these models with 3D reconstruction, similar to [17, 18], enables efficient and high quality 3D creation. Below we discuss how our work is related to several areas of prior work.

**2D priors.** Given limited information such as text, pretrained text-to-image models can provide a strong generative prior for text-to-3D generation. However, distilling the knowledge present in these image-based priors into a coherent 3D model currently requires an iterative distillation approach. DreamFusion [8] introduced Score Distillation Sampling (SDS) to synthesize 3D objects (as NeRFs) from text prompts. Research in this space has aimed to improve distillation strategies [19, 20, 21, 22,

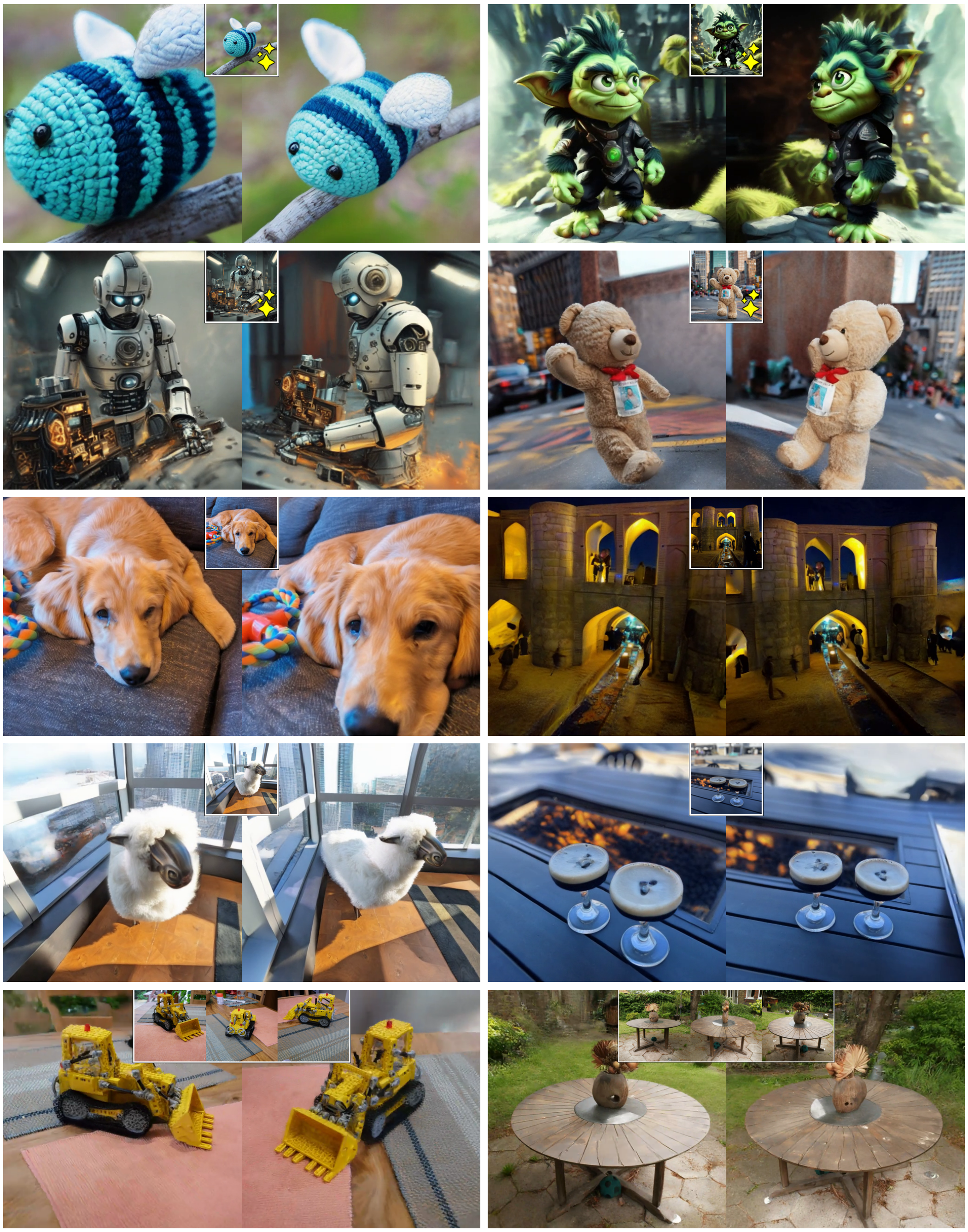

Figure 2: **Qualitative results (renders)**: CAT3D can create high-quality 3D objects or scenes from a number of input modalities: an input image generated by a text-to-image model (*top row*), a single captured real image (*middle row*), and multiple captured real images (*bottom row*).

23], swap in other 3D representations [24, 25, 26, 27, 28], and amortize the optimization process [29]. Using text-based priors for single-image-to-3D has also shown promise [30, 31, 32], but requires a complex balancing of the image observation with additional constraints. Incorporating priors such as monocular depth models or inpainting models has been useful for creating 3D scenes, but tends to result in poor global geometry [33, 34, 35, 36].

**2D priors with camera conditioning.** While text-to-image models excel at generating visually appealing images, they lack precise control over the pose of images, and thus require a time-consuming 3D distillation process to encourage the 3D model to conform to the 2D prior. To overcome this limitation, several approaches train or fine-tune generative models with explicit image and pose conditioning [7, 37, 38, 39, 40, 41, 42]. These models provide stronger priors for what an object or scene should look like given text and/or input image(s), but they also model all output views *independently*. In cases where there is little uncertainty in what should appear at novel views, reasoning about generated views independently is sufficient for efficient 3D reconstruction [43]. But when there is some uncertainty exists, these top-performing methods still require expensive 3D distillation to resolve the inconsistencies between different novel views.

**Multi-view priors.** Modeling the correlations between multiple views provides a much stronger prior for what 3D content is consistent with partial observations. Methods like MVDream [44], ImageDream [9], Zero123++ [45], ConsistNet [46], SyncDreamer [47] and ViewDiff [48] fine-tune text-to-image models to generate multiple views simultaneously. CAT3D is similar in architecture to ImageDream, where the multi-view dependency is captured by an architecture resembling video diffusion models with 3D self-attention. Given this stronger prior, these papers also demonstrate higher quality and more efficient 3D extraction.

**Video priors.** Video diffusion models have demonstrated an astonishing capability of generating realistic videos [49, 50, 10, 12, 15, 13, 51], and are thought to implicitly reason about 3D. However, it remains challenging to use off-the-shelf video diffusion models for 3D generation for a number of reasons. Current models lack exact camera controls, limiting generation to clips with only smooth and short camera trajectories, and struggle to generate videos with only camera motion but no scene dynamics. Several works have proposed to resolve these challenges by fine-tuning video diffusion models for camera-controled or multi-view generation. For example, AnimateDiff [52] LoRA fine-tuned a video diffusion model with fixed types of camera motions, and MotionCtrl [53] conditioned the model on arbitrary specified camera trajectories. ViVid-1-to-3 [54] combines a novel view synthesis model and a video diffusion model for generating smooth trajectories. SVD-MV [55], IM-3D [17] and SV3D [55] further explored leveraging camera-controlled or multi-view video diffusion models for 3D generation. However, their camera trajectories are limited to orbital ones surrounding the center content. These approaches mainly focus on 3D object generation, and do not work for 3D scenes, few-view 3D reconstruction, or objects in context (objects that have not been masked or otherwise separated from the image's background).

**Feed-forward methods.** Another line of research is to learn feed-forward models that take a few views as input, and output 3D representations directly, without an optimization process per instance [6, 56, 57, 58, 18, 59, 60]. These methods can produce 3D representations efficiently (within a few seconds), but the quality is often worse than approaches built on image-space priors.

## 3 Method

CAT3D is a two-step approach for 3D creation: first, we generate a large number of novel views consistent with one or more input views using a multi-view diffusion model, and second, we run a robust 3D reconstruction pipeline on the generated views (see Figure 3). Below we describe our multi-view diffusion model (Section 3.1), our method for generating a large set of nearly consistent novel views from it (Section 3.2), and how these generated views are used in a 3D reconstruction pipeline (Section 3.3).

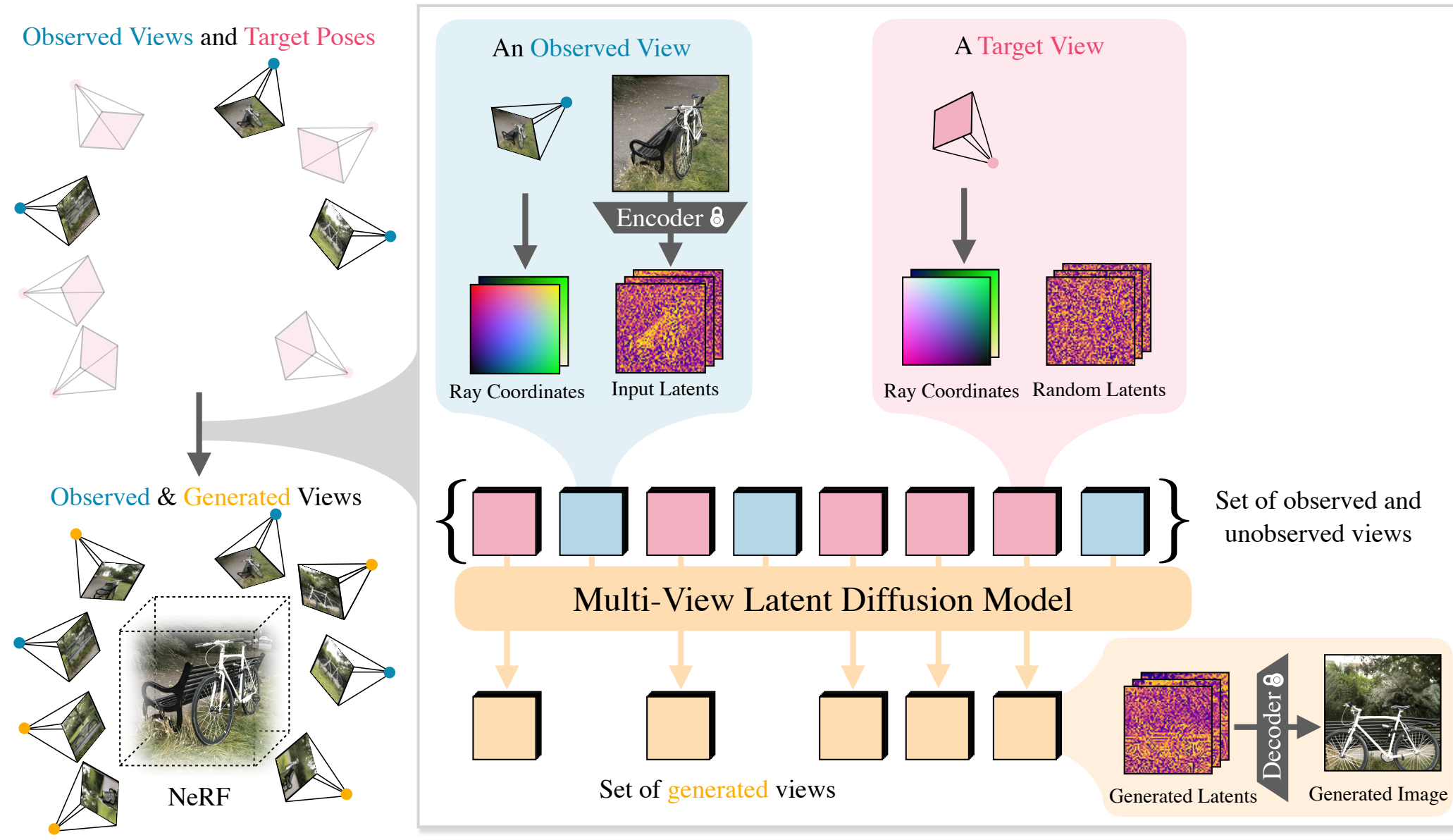

Figure 3: **Illustration of the method.** Given one to many views, CAT3D creates a 3D representation of the scene in as little as one minute. CAT3D has two stages: (1) generate a large set of synthetic views with a multi-view latent diffusion model conditioned on the input views and target poses; (2) run a robust 3D reconstruction pipeline on the observed and generated views. This decoupling of the generative prior and 3D reconstruction process results in efficiency improvements and reduced methodological complexity relative to prior work [7, 8, 42], while also improving visual quality.

### 3.1 Multi-View Diffusion Model

We train a multi-view diffusion model that takes a single or multiple views of a 3D scene as input and generates multiple output images given their camera poses (where "a view" is a paired image and its camera pose). Specifically, given $M$ conditional views containing $M$ images $\mathbf{I}^{\text{cond}}$ and their corresponding camera parameters $\mathbf{p}^{\text{cond}}$, the model learns to capture the joint distribution of $N$ target images $\mathbf{I}^{\text{tgt}}$ assuming their $N$ target camera parameters $\mathbf{p}^{\text{tgt}}$ are also given:

$$p\left(\mathbf{I}^{\text{tgt}} | \mathbf{I}^{\text{cond}}, \mathbf{p}^{\text{cond}}, \mathbf{p}^{\text{tgt}}\right). \tag{1}$$

**Model architecture.** Our model architecture is similar to video latent diffusion models (LDMs) [49, 11], but with camera pose embeddings for each image instead of time embeddings. Given a set of conditional and target images, the model encodes every individual image into a latent representation through an image variational auto-encoder [61]. Then, a diffusion model is trained to estimate the joint distribution of the latent representations given conditioning signals. We initialize the model from an LDM trained for text-to-image generation similar to [62] trained on web-scale image data, with an input image resolution of $512 \times 512 \times 3$ and latents with shape $64 \times 64 \times 8$. As is often done in video diffusion models [50, 10, 11], the main backbone of our model remains the pretrained 2D diffusion model but with additional layers connecting the latents of multiple input images. As in [44], we use 3D self-attention (2D in space and 1D across images) instead of simple 1D self-attention across images. We directly inflate the existing 2D self-attention layers after every 2D residual block of the original LDM to connect latents with 3D self-attention layers while inheriting the parameters from the pre-trained model, introducing minimal amount of extra model parameters. We found that conditioning on input views through 3D self-attention layers removed the need for PixelNeRF [63] and CLIP image embeddings [64] used by the prior state-of-the-art model on few-view reconstruction, ReconFusion [7]. We use FlashAttention [65, 66] for fast training and sampling, and fine-tune all the weights of the latent diffusion model. Similar to prior work [10, 67], we found it important to shift the noise schedule towards high noise levels as we move from the pre-trained image diffusion model to our multi-view diffusion model that captures data of higher dimensionality. Concretely, following logic similar to [67], we shift the log signal-to-noise ratio by $\log(N)$ towards the high signal-to-noise ratio region, where $N$ is the number of target views. Similar shifts have been adopted empirically

by [11, 10]. For training, latents of target images are noise perturbed while latents of conditional images are kept as clean, and the diffusion loss is defined only on target images. A binary mask is concatenated to the latents along the channel dimension, to denote conditioning vs. target images. To deal with multiple 3D generation settings, we train a single versatile model that can model a total of 8 conditioning and target views ($N + M = 8$), and randomly select the number of conditional views $M$ to be $1$ or $3$ during training, corresponding to 7 and 5 target views respectively. The noise schedule is shifted based on $5$ target views. See Appendix B for more model details.

**Camera conditioning.** To condition on the camera pose, we use a camera ray representation ("raymap") that is the same height and width as the latent representations [38, 68] and encodes the ray origin and direction at each spatial location. The rays are computed relative to the camera pose of the first conditional image, so our pose representation is invariant to rigid transformations of 3D world coordinates. Raymaps for each image are concatenated channel-wise onto the latents for the corresponding image.

### 3.2 Generating Novel Views

Given a set of input views, our goal is to generate a large set of consistent views to fully cover the scene and enable accurate 3D reconstruction. To do this, we need to decide on the set of camera poses to sample, and we need to design a sampling strategy that can use a multi-view diffusion model trained on a small number of views to generate a much larger set of consistent views.

**Camera trajectories.** Compared to 3D object reconstruction where orbital camera trajectories can be effective, a challenge of 3D scene reconstruction is that the views required to fully cover a scene can be complex and depend on the scene content. We empirically found that designing reasonable camera trajectories for different types of scenes is crucial to achieve compelling few-view 3D reconstruction. The camera paths must be sufficiently thorough and dense to fully-constrain the reconstruction problem, but also must not pass through objects in the scene or view scene content from unusual angles. In summary, we explore four types of camera paths based on the characteristic of a scene: (1) orbital paths of different scales and heights around the center scene, (2) forward facing circle paths of different scales and offsets, (3) spline paths of different offsets, and (4) spiral trajectories along a cylindrical path, moving into and out of the scene. See Appendix C for more details.

**Generating a large set of synthetic views.** A challenge in applying our multi-view diffusion model to novel view synthesis is that it was trained with a small and finite set of input and output views — just 8 in total. To increase the total number of output views, we cluster the target viewpoints into smaller groups and generate each group independently given the conditioning views. We group target views with close camera positions, as these views are typically the most dependent. For single-image conditioning, we adopt an autoregressive sampling strategy, where we first generate a set of 7 *anchor views* that cover the scene (similar to [42], and chosen using the greedy initialization from [69]), and then generate the remaining groups of views in parallel given the observed and anchor views. This allows us to efficiently generate a large set of synthetic views while still preserving both long-range consistency between anchor views and local similarity between nearby views. For the single-image setting, we generate 80 views, while for the few-view setting we use 480-960 views. See Appendix C for details.

**Conditioning larger sets of input views and non-square images.** To expand the number of views we can condition on, we choose the nearest $M$ views as the conditioning set, as in [7]. We experimented with simply increasing the sequence length of the multi-view diffusion architecture during sampling, but found that the nearest view conditioning and grouped sampling strategy performed better. To handle wide aspect ratio images, we combine square samples from square-cropped input views with wide samples cropped from input views padded to be square.

### 3.3 Robust 3D reconstruction

Our multi-view diffusion model generates a set of high-quality synthetic views that are reasonably consistent with each other. However, the generated views are generally not perfectly 3D consistent. Indeed, generating perfectly 3D consistent images remains a very challenging problem even for

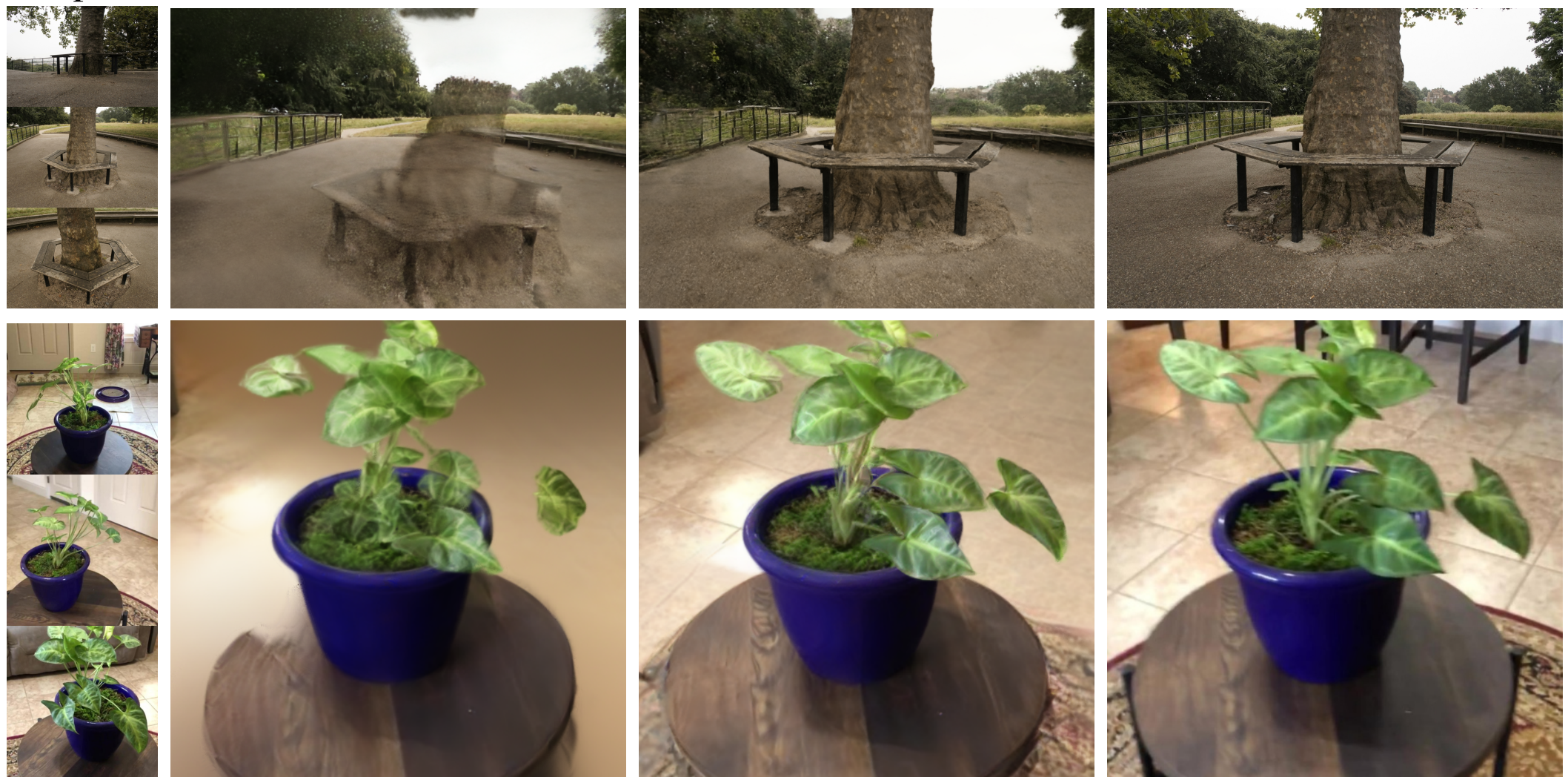

Figure 4: **Qualitative comparison, few-view reconstruction (renders).** A comparison of rendered reconstructions on scenes from mip-NeRF 36 (*top row*) and CO3D (*bottom row*), given 3 input captured views. Compared to ReconFusion [7], CAT3D better aligns with ground-truth in seen regions, while hallucinating plausible content in unseen regions. See supplemental website for additional comparisons.

current state-of-the-art video diffusion models [70]. Since 3D reconstruction methods have been designed to take photographs (which are by definition perfectly consistent) as input, we modify the standard NeRF training procedure to improve its robustness to inconsistent input views.

We build upon Zip-NeRF [71], whose training procedure minimizes the sum of a photometric reconstruction loss, a distortion loss, an interlevel loss, and a normalized L2 weight regularizer. We additionally include a perceptual loss (LPIPS [72]) between the rendered image and the input image. Compared to the photometric reconstruction loss, LPIPS emphasizes high-level semantic similarity between the rendered and observed images, while ignoring potential inconsistencies in low-level high-frequency details. Since generated views closer to the observed views tend to have less uncertainty and are therefore more consistent, we weight the losses for generated views based on the distance to the nearest observed view. This weighting is uniform at the beginning of the training, and is gradually annealed to a weighting function that more strongly penalizes reconstruction losses for views closer to one of the observed views. See Appendix D for additional details.

## 4 Experiments

We trained the multi-view diffusion model at the core of CAT3D on four datasets with camera pose annotations: Objaverse [73], CO3D [74], RealEstate10k [75] and MVImgNet [76]. We then evaluated CAT3D on the few-view reconstruction task (Section 4.1) and the single image to 3D task (Section 4.2), demonstrating qualitative and quantitative improvements over prior work. The design choices that led to CAT3D are ablated and discussed further in Section 4.3.

### 4.1 Few-View 3D Reconstruction

We first evaluate CAT3D on five real-world benchmark datasets for few-view 3D reconstruction. Among those, CO3D [74] and RealEstate10K [75] are in-distribution datasets whose training splits were part of our training set (we use their test splits for evaluation), whereas DTU [77], LLFF [78] and the mip-NeRF 360 dataset [79] are out-of-distribution datasets that were not part of the training dataset. We tested CAT3D on the 3, 6 and 9 view reconstruction tasks, with the same train and eval splits as [7]. In Table 1, we compare to the state-of-the-art for dense-view NeRF reconstruction with no learned priors (Zip-NeRF [71]) and methods that heavily leverage generative priors such as ZeroNVS [42]

| Dataset | 3-view | | | 6-view | | | 9-view | | |
|---|---|---|---|---|---|---|---|---|---|
| Method | PSNR ↑ | SSIM ↑ | LPIPS ↓ | PSNR ↑ | SSIM ↑ | LPIPS ↓ | PSNR ↑ | SSIM ↑ | LPIPS ↓ |
| **RealEstate10K** | | | | | | | | | |
| Zip-NeRF* [71] | 20.77 | 0.774 | 0.332 | 27.34 | 0.906 | 0.180 | 31.56 | 0.947 | 0.118 |
| ZeroNVS* [42] | 19.11 | 0.675 | 0.422 | 22.54 | 0.744 | 0.374 | 23.73 | 0.766 | 0.358 |
| ReconFusion [7] | 25.84 | 0.910 | 0.144 | 29.99 | 0.951 | 0.103 | 31.82 | 0.961 | 0.092 |
| CAT3D (ours) | 26.78 | 0.917 | 0.132 | 31.07 | 0.954 | 0.092 | 32.20 | 0.963 | 0.082 |
| **LLFF** | | | | | | | | | |
| Zip-NeRF* [71] | 17.23 | 0.574 | 0.373 | 20.71 | 0.764 | 0.221 | 23.63 | 0.830 | 0.166 |
| ZeroNVS* [42] | 15.91 | 0.359 | 0.512 | 18.39 | 0.449 | 0.438 | 18.79 | 0.470 | 0.416 |
| ReconFusion [7] | 21.34 | 0.724 | 0.203 | 24.25 | 0.815 | 0.152 | 25.21 | 0.848 | 0.134 |
| CAT3D (ours) | 21.58 | 0.731 | 0.181 | 24.71 | 0.833 | 0.121 | 25.63 | 0.860 | 0.107 |
| **DTU** | | | | | | | | | |
| Zip-NeRF* [71] | 9.18 | 0.601 | 0.383 | 8.84 | 0.589 | 0.370 | 9.23 | 0.592 | 0.364 |
| ZeroNVS* [42] | 16.71 | 0.716 | 0.223 | 17.70 | 0.737 | 0.205 | 17.92 | 0.745 | 0.200 |
| ReconFusion [7] | 20.74 | 0.875 | 0.124 | 23.62 | 0.904 | 0.105 | 24.62 | 0.921 | 0.094 |
| CAT3D (ours) | 22.02 | 0.844 | 0.121 | 24.28 | 0.899 | 0.095 | 25.92 | 0.928 | 0.073 |
| **CO3D** | | | | | | | | | |
| Zip-NeRF* [71] | 14.34 | 0.496 | 0.652 | 14.48 | 0.497 | 0.617 | 14.97 | 0.514 | 0.590 |
| ZeroNVS* [42] | 17.13 | 0.581 | 0.566 | 19.72 | 0.627 | 0.515 | 20.50 | 0.640 | 0.500 |
| ReconFusion [7] | 19.59 | 0.662 | 0.398 | 21.84 | 0.714 | 0.342 | 22.95 | 0.736 | 0.318 |
| CAT3D (ours) | 20.57 | 0.666 | 0.351 | 22.79 | 0.726 | 0.292 | 23.58 | 0.752 | 0.273 |
| **Mip-NeRF 360** | | | | | | | | | |
| Zip-NeRF* [71] | 12.77 | 0.271 | 0.705 | 13.61 | 0.284 | 0.663 | 14.30 | 0.312 | 0.633 |
| ZeroNVS* [42] | 14.44 | 0.316 | 0.680 | 15.51 | 0.337 | 0.663 | 15.99 | 0.350 | 0.655 |
| ReconFusion [7] | 15.50 | 0.358 | 0.585 | 16.93 | 0.401 | 0.544 | 18.19 | 0.432 | 0.511 |
| CAT3D (ours) | 16.62 | 0.377 | 0.515 | 17.72 | 0.425 | 0.482 | 18.67 | 0.460 | 0.460 |

Easier ↑ Dataset Difficulty ↓ Harder

Table 1: **Quantitative comparison of few-view 3D reconstruction**. CAT3D outperforms baseline approaches across nearly all settings and metrics (modified baselines denoted with * taken from [7]).

and ReconFusion [7]. We find that CAT3D achieves state-of-the-art performance across nearly all settings, while also reducing generation time from 1 hour (for ZeroNVS and ReconFusion) down to a few minutes. CAT3D outperforms baseline approaches on more challenging datasets like CO3D and mip-NeRF 360 by a larger margin, thereby demonstrating its value in reconstructing large and highly detailed scenes. Figure 4 shows the qualitative comparison. In unobserved regions, CAT3D is able to hallucinate plausible textured content while still preserving geometry and appearance from the input views, whereas prior works often produce blurry details and oversmoothed backgrounds.

### 4.2 Single image to 3D

CAT3D supports the efficient generation of diverse 3D content from just a single input view. Evaluation in this under-constrained regime is challenging as there are many 3D scenes consistent with the single view, for example scenes of different scales. We thus focus our single image evaluation on qualitative comparisons (Figure 5), and quantitative semantic evaluations with CLIP [64] (Table 2). On scenes, CAT3D produces higher resolution results than ZeroNVS [42] and RealmDreamer [80], and for both scenes and objects we better preserve details from the input image. On images with segmented objects, our geometry is often worse than existing approaches like ImageDream [9] and DreamCraft3D [81], but maintains competitive CLIP scores. Compared to these prior approaches that iteratively leverage a generative prior in 3D distillation, CAT3D is more than an order of magnitude faster. Faster generation methods have been proposed for objects [6, 82, 83], but produce significantly lower resolution results than their iterative counterparts, so they are not included in this comparison. IM-3D [17] achieves better performance on segmented objects with similar runtime, but does not work on scenes, or on objects in context.

### 4.3 Ablations

At the core of CAT3D is a multi-view diffusion model that has been trained to generate consistent novel views. We considered several model variants, and evaluated both their sample quality (on

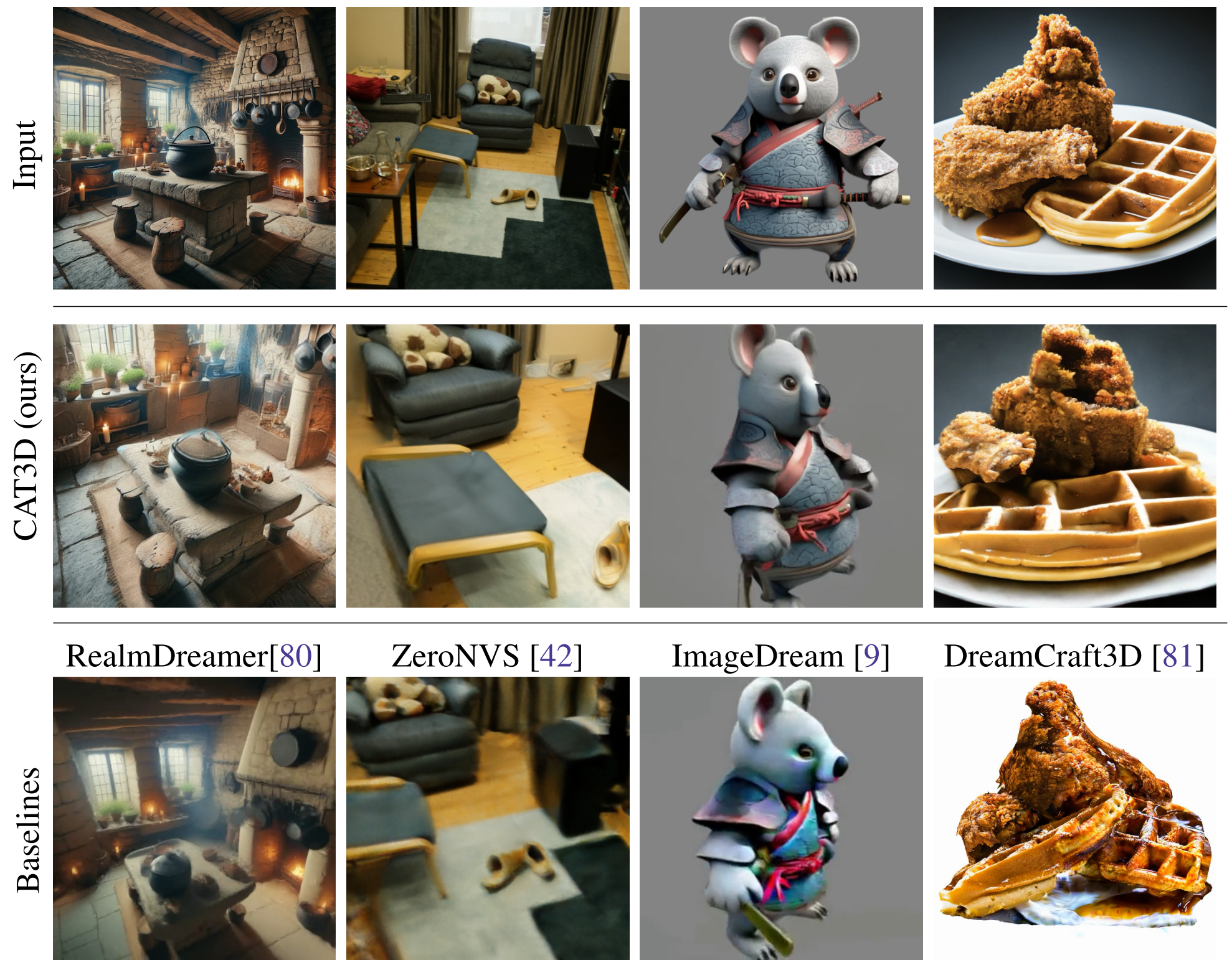

Figure 5: **3D creation from single input images**. Renderings of 3D models from CAT3D (*middle row*) are higher quality than baselines (*bottom row*) for scenes, and competitive for objects. Note that scale ambiguity amplifies the differences in renderings between methods. See supplemental website for additional comparisons.

| Model | Time (min) | CLIP (Image) |
|---|---|---|
| ImageDream [9] | 120 | 83.77 ± 5.2 |
| One2345++ [84] | 0.75 | 83.78 ± 6.4 |
| IM-3D (NeRF) [17] | 40 | 87.37 ± 5.4 |
| IM-3D [17] | 3 | **91.40** ± 5.5 |
| CAT3D (ours) | 1 | 88.54 ± 8.6 |

Table 2: Evaluating image-to-3D quality with CLIP image scores on examples from [9] (numbers reproduced from [17]). CAT3D produces competitive results to object-centric baselines while also working on whole scenes. (Note: shown images are 3D renders, and time for CAT3D was evaluated on 16 A100 GPUs.)

in-domain and out-of-domain datasets) and few-view 3D reconstruction performance. We also compare important design choices for 3D reconstruction. Results from our ablation study are reported in Table 3 and Figure 6 in Appendix A and summarized below. Overall, we found that video diffusion architectures, with 3D self-attention (spatiotemporal) and raymap embeddings of camera pose, produce consistent enough views to recover 3D representations when combined with robust reconstruction losses.

**Image and pose.** Previous work [7] used PixelNerf [63] feature-map conditioning for multiple input views. We found that replacing PixelNeRF with attention-based conditioning in a conditional video diffusion architecture using a per-image embedding of the camera pose results in improved samples and 3D reconstructions, while also reducing model complexity and the number of parameters. We found that embedding the camera pose as a low-dimensional vector (as in [37]) works well for in-domain samples, but generalizes poorly compared to raymap conditioning (see Section 3.1).

**Increasing the number of views.** We found that jointly modeling multiple output views (*i.e.*, 5 or 7 views instead of 1) improves sample metrics — even metrics that evaluate the quality of each output image *independently*. Jointly modeling multiple outputs creates more consistent views that result in an improved 3D reconstruction as well.

**Attention layers.** We found that 3D self-attention (spatiotemporal) is crucial, as it yields improved performance relative to factorized 2D self-attention (spatial-only) and 1D self-attention (temporal-only). While models with 3D self-attention in the finest feature maps ($64 \times 64$) result in the highest fidelity images, they incur a significant computational overhead for training and sampling for relative small gain in fidelity. We therefore decided to use 3D self-attention only in feature maps of size $32 \times 32$ and smaller.

**Multi-view diffusion model training.** Initializing from a pre-trained text-to-image latent diffusion model improved performance on out-of-domain examples. We experimented with fine-tuning the multi-view diffusion model to multiple variants specialized for specific numbers of inputs and outputs views, but found that a single model jointly trained on 8 frames with either 1 or 3 conditioning views was sufficient to enable accurate single image and few-view 3D reconstruction.

**3D reconstruction.** LPIPS loss is crucial for achieving high-quality texture and geometry, aligning with findings in [17, 7]. On Mip-NeRF 360, increasing the number of generated views from 80 (single elliptical orbit) to 720 (nine orbits) improved central object geometry but sometimes introduced background blur, probably due to inconsistencies in generated content.

## 5 Discussion

We present CAT3D, a unified approach for 3D content creation from any number of input images. CAT3D leverages a multi-view diffusion model for generating highly consistent novel views of a 3D scene, which are then input into a 3D multi-view reconstruction pipeline. CAT3D decouples the generative prior from 3D extraction, leading to efficient, simple, and high-quality 3D generation.

Although CAT3D produces compelling results and outperforms prior works on multiple tasks, it has limitations. Because our training datasets have roughly constant camera intrinsics for views of the same scene, the trained model cannot handle test cases well where input views are captured by multiple cameras with different intrinsics. The generation quality of CAT3D relies on the expressivity of the base text-to-image model, and it performs worse in the cases where scene content is out of distribution for the base model (e.g. human faces, since the base model was trained on limited human data). The number of output views supported by our multi-view diffusion model is still relatively small, so when we generate a large set of samples from our model, not all views may be 3D consistent with each other (see Supplementary website). Finally, CAT3D uses manually-constructed camera trajectories that cover the scene thoroughly (see Appendix C), which may be difficult to design for large-scale open-ended 3D environments.

There are a few directions worth exploring in future work to improve CAT3D. The multi-view diffusion model may benefit from being initialized from a pre-trained video diffusion model, as observed by [10, 17]. The consistency of samples could be further improved by extending the number of conditioning and target views handled by the model. Automatically determining the camera trajectories required for different scenes could increase the flexibility of the system.

**Acknowledgements** We would like to thank Daniel Watson, Rundi Wu, Jason Y. Zhang, Richard Tucker, Jason Baldridge, Michael Niemeyer, Rick Szeliski, Dana Roth, Jordi Pont-Tuset, Andeep Torr, Irina Blok, Doug Eck, and Henna Nandwani for their valuable contributions to this work. We also extend our gratitude to Shlomi Fruchter, Kevin Murphy, Mohammad Babaeizadeh, Han Zhang and Amir Hertz for training the base text-to-image latent diffusion model.

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

# A Ablations

Here we conduct ablation studies over several important decisions that led to CAT3D. We consider several multi-view diffusion model variants, and evaluated them on novel-view synthesis with held-out validation sets (4k samples) from the training datasets (*in-domain samples*) and from the mip-NeRF 360 dataset (*out-of-domain samples*), as well as 3-view reconstruction on the mip-NeRF 360 dataset (*out-of-domain renders*). Unless otherwise specified, models in this section are trained with 3 input views and 5 output views, evaluated after 120k optimization iterations. Quantitative results are summarized in Table 3. Then we discuss important 3D reconstruction design choices, with qualitative comparison in Figure 6.

**Number of target views.** We start from the setting where the model takes 3 conditional views as input and generates a single target view, identical to the ReconFusion baseline [7]. Results show that our multi-view architecture is favored when dealing with multiple input views, compared to the PixelNeRF used by [7]. We also show that going from a single target view to 5 target views results in a significant improvement on both novel-view synthesis and few-view reconstruction.

**Camera conditioning.** As mentioned in Section 3.1, we compared two camera parameterizations, one is an 8-dimensional encoding vector fed through cross-attention layers that contains relative position, relative rotation quaternion, and absolute focal length. The other is camera rays fed through channel-wise concatenation with the input latents. We observe that the latter performs better across all metrics.

**Attention layers.** An important design choice is the type and number of self-attention layers used when connecting multiple views. As shown in Table 3, it is critical to use 3D self-attention instead of temporal 1D self-attention. However, 3D attention is expensive; 3D attention at the largest feature maps ($64 \times 64$) is of 32k sequence length and this incurs a significant computation overhead during training and sampling with a marginal performance gain, especially for out-of-domain samples and renderings. We therefore chose to use 3D attention only for feature maps of size $32 \times 32$ and smaller.

**Multi-view diffusion model training.** We compared the settings of training the multi-view diffusion models from scratch with initializing from a pre-trained text-to-image latent diffusion model. The latter performs better, especially for out-of-domain cases. We further trained the model for more iterations and observed a consistent performance gain up until 1M iterations. Then we fine-tuned the model for handling both cases of 1 conditional + 7 target views and 3 conditional + 5 target views (*jointly*), for another $0.4$M iterations. We found this joint finetuning leads to better in-domain novel-view synthesis results with 3 conditional views, and out-of-domain results that are on-par with the previous model.

**3D reconstruction.** We found perceptual distance (LPIPS) loss is crucial in recovering high-quality texture and geometry, similar to [17, 7]. We also compared the use of 80 views along one orbital path with 720 views along nine variably scaled orbital paths. In the Mip-NeRF 360 setting, increasing the number of views helps better regularize the geometry of central objects, but sometimes leads to blurrier textures in the background, due to inconsistencies in generated content.

# B Details of Multi-View Diffusion Model

We initialize the multi-view diffusion model from a latent diffusion model (LDM) trained for text-to-image generation similar to [62] trained on web scale image datasets. See Figure 7 for a visualization of the model architecture. We modify the LDM to take multi-view images as input by inflating the 2D self-attention after every 2D residual blocks to 3D self-attention [44]. Our model adds minimal additional parameters to the backbone model: just a few additional convolution channels at the input layer to handle conditioning information. We drop the text embedding from the original model. Our latent diffusion model has 850M parameters, smaller than existing approaches built on video diffusion models such as IM-3D [17] (4.3B) and SV3D [55] (1.5B).

We fine-tune the full latent diffusion model for 1.4M iterations with a batch size of 128 and a learning rate of $5 \times 10^{-5}$. The first 1M iterations are trained with the setting of 1 conditional view and 7

| Setting | In-domain diffusion samples PSNR↑ | SSIM↑ | LPIPS↓ | Out-of-domain diffusion samples PSNR↑ | SSIM↑ | LPIPS↓ | Out-of-domain NeRF renderings PSNR↑ | SSIM↑ | LPIPS↓ |
|---|---|---|---|---|---|---|---|---|---|
| **Baseline** | | | | | | | | | |
| ReconFusion [7] | — | — | — | 14.01 | 0.265 | 0.568 | 15.49 | 0.358 | 0.585 |
| **# target views** | | | | | | | | | |
| 3 cond 1 tgt | 18.85 | 0.638 | 0.359 | 14.12 | 0.262 | 0.553 | 16.17 | 0.360 | 0.546 |
| 3 cond 5 tgt | **21.66** | **0.733** | **0.277** | **14.63** | **0.278** | **0.515** | **16.29** | **0.368** | **0.530** |
| **Camera conditioning** | | | | | | | | | |
| Low-dim vector | 21.17 | 0.710 | 0.304 | 14.19 | 0.266 | 0.530 | 15.97 | 0.359 | 0.544 |
| Raymap | **21.66** | **0.733** | **0.277** | **14.63** | **0.278** | **0.515** | **16.29** | **0.368** | **0.530** |
| **Attention layers** | | | | | | | | | |
| Temporal attention | 18.62 | 0.653 | 0.362 | 13.41 | 0.250 | 0.582 | 15.03 | 0.330 | 0.595 |
| 3D attention until $16 \times 16$ | 21.41 | 0.730 | 0.281 | 14.23 | 0.274 | 0.530 | 16.21 | 0.364 | 0.541 |
| 3D attention until $32 \times 32$ | 21.66 | 0.733 | 0.277 | 14.63 | **0.278** | 0.515 | 16.29 | **0.368** | 0.530 |
| 3D attention until $64 \times 64$ (full) | **22.83** | **0.783** | **0.235** | **14.64** | 0.274 | **0.509** | **16.35** | 0.367 | **0.528** |
| **Model training** | | | | | | | | | |
| From scratch, 3 cond 5 tgt | 21.16 | 0.722 | 0.282 | 13.88 | 0.255 | 0.546 | 15.68 | 0.348 | 0.557 |
| From pretrained, 3 cond 5 tgt | 21.66 | 0.733 | 0.277 | 14.63 | 0.278 | 0.515 | 16.29 | 0.368 | 0.530 |
| From pretrained, 3 cond 5 tgt, 1M iters | 22.49 | 0.757 | 0.256 | **15.19** | **0.303** | **0.482** | 16.58 | **0.384** | **0.509** |
| From pretrained, jointly, 1.4M iters | **22.96** | **0.777** | **0.235** | 15.15 | 0.294 | 0.488 | **16.62** | 0.377 | 0.515 |

Table 3: **Ablation study of multi-view diffusion models.** A comparison of model variants and parameters settings across *in-domain* sequences (a mixture of Objaverse, Co3D, RealEstate10k and MVImgNet — all sequences in the corresponding eval sets of our training data) and *out-of-domain* sequences (examples from the mip-NeRF 360 dataset). We evaluate the quality of *samples* from the diffusion model, as well as *renderings* from the subsequently optimized NeRF.

target views, while the rest $0.4$M iterations are trained with an equal mixture of 1 condition + 7 target views and 3 conditional + 5 target views. Our model was trained for 16 days on 128 TPU-v4 chips. Following [7] we draw training samples with equal probability from the four training datasets. We enable classifier-free guidance (CFG) [85] by randomly dropping the conditional images and camera poses with a probability of $0.1$ during training.

## C Details of Generating Novel Views

We use DDIM [86] with $50$ sampling steps and CFG guidance weight 3 for generating novel views. It takes 5 seconds to generate $80$ views on 16 A100 GPUs. As mentioned in Section 3.2, selecting camera trajectories that fully cover the 3D scene is important for high-quality 3D generation results. See Figure 8 for an illustration of the camera trajectories we use. For the single image-to-3D setting we use two different types of camera trajectories, each containing $80$ views:

- A spiral around a cylinder-like trajectory that moves into and out of the scene.
- An orbit trajectory for images with a central object.

For few-view reconstruction, we create different trajectories based on the characteristics of different datasets:

- RealEstate10K: we create a spline path fitted from the input views, and shift the trajectories along the $xz$-plane by certain offsets, resulting in 800 views.
- LLFF and DTU: we create a forward-facing circle path fitted from all views in the training set, scale it, and shift along the $z$-axis by certain offsets, resulting in 960 and 480 views respectively.
- CO3D: we create a spline path fitted from the input views, and scale the trajectories by multiple factors, resulting in 640 views.
- Mip-NeRF 360: we create a elliptical path fitted from all views in the training set , scale it, and shift along the $z$-axis by certain offsets, resulting in 720 views.

For generating anchor views in the single-image setting, we used the model with 1 conditional input and 7 target outputs. For generating the full set of views (both in the single-image anchored setting,

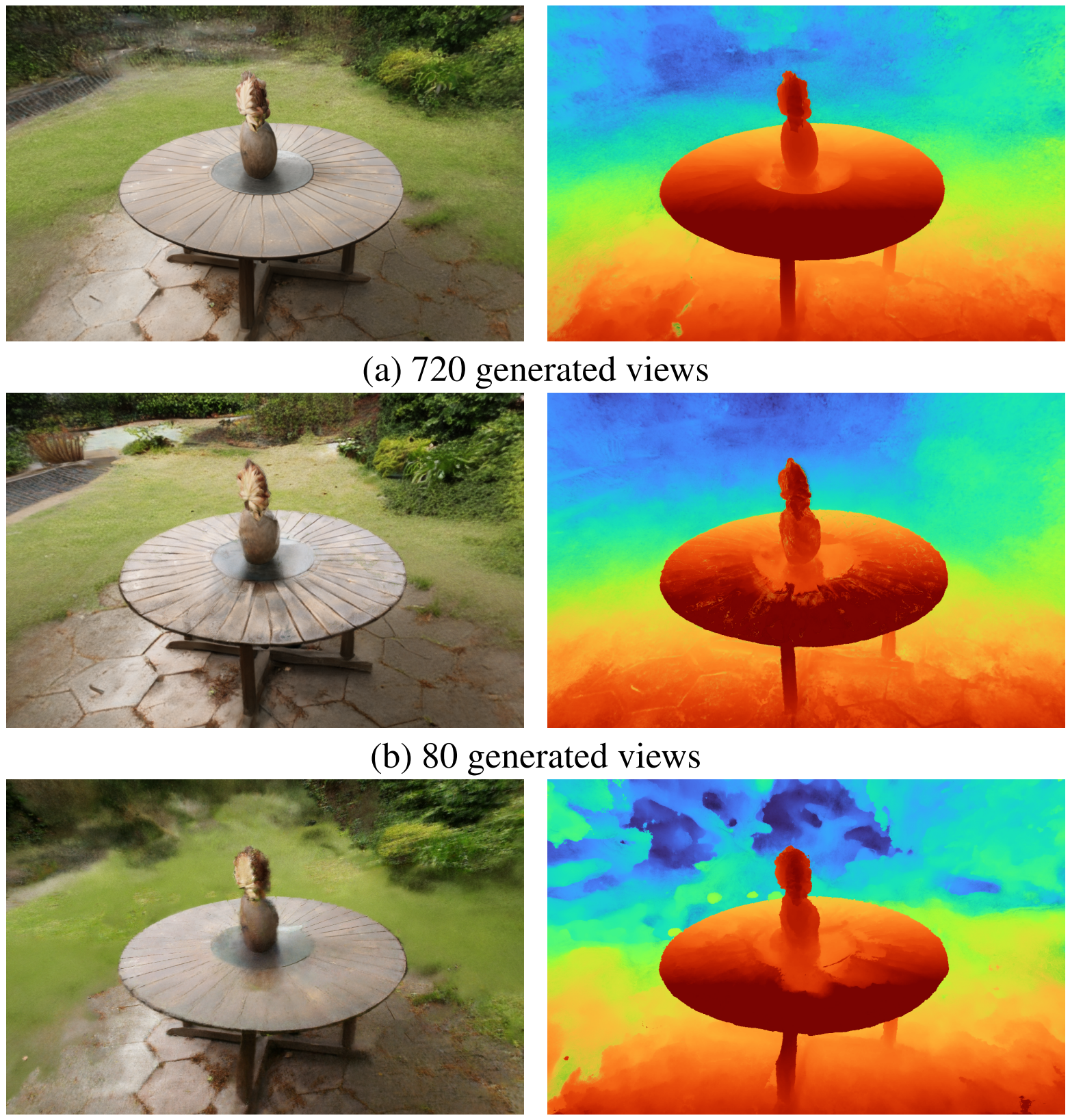

(a) 720 generated views

(b) 80 generated views

(c) Without LPIPS loss

Figure 6: **Qualitative comparison of 3D reconstruction design choices.** Rendered images (left) and depth maps (right) of a Mip-NeRF 360 scene under different settings: (a) 720 generated views along multiple orbital paths, (b) 80 generated views on a single orbital path, and (c) 720 views, without the perceptual (LPIPS) loss.

as well as in the multi-view setting), we used the model with 3 conditional inputs and groups of 5 target outputs, selected by their indices in the camera trajectories.

**Anchor selection.** For single-image conditioning, we first select a set of target viewpoints as *anchors* to ground the scene content. To select a set of anchor views that are spread throughout the scene and provide good coverage, we use the initialization strategy from [69]. This method greedily selects the camera whose position is furthest away from the already selected views. We found this to work well on a variety of trajectory-like view sets as well as random views that have been spread throughout a scene.

**Dealing with non-square images.** While the multi-view latent diffusion model we trained only supports $512 \times 512$ square images, we found the model still performed well when padding non-square images to square. However, this method often reduces resolution, so we also run our model on a square-cropped version of the inputs, and then compose the square-cropped outputs with the edges from the padded outputs to create a different aspect ratio image.

## D Details of 3D Reconstruction

For the Zip-NeRF [71] baseline, we follow [7] to make a few modifications of the hyperparameters (See Appendix D in [7]) that better suit the few-view reconstruction setting. We use a smaller view dependence network with width 32 and depth 1, and a smaller number of training iterations of 1000, which helps avoid overfitting and substantially speeds up the training and rendering. Our synthetic view sampling and 3D reconstruction process is run on 16 A100 GPUs. For few-view reconstruction,

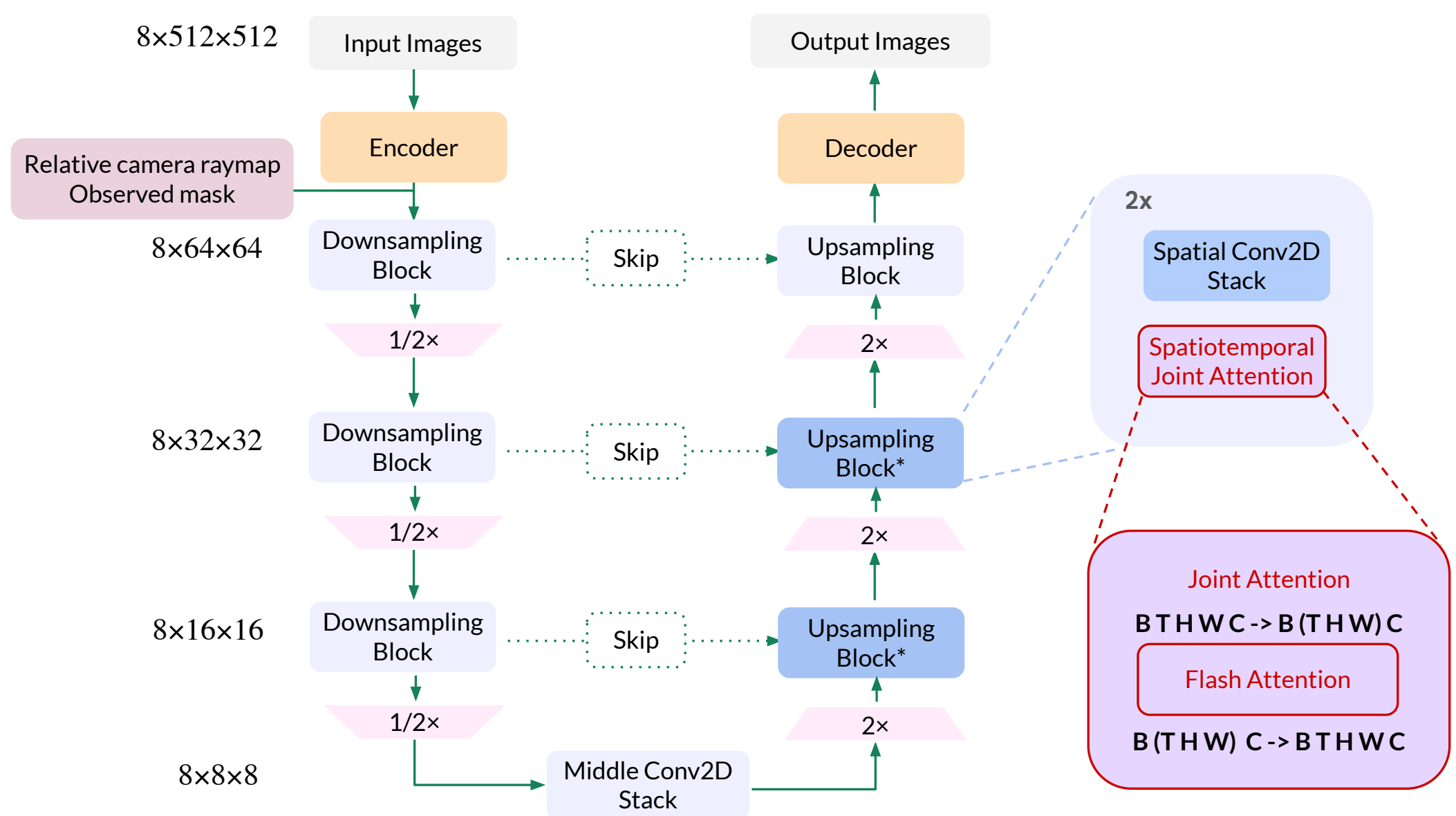

Figure 7: **Illustration of the network.** CAT3D builds on a latent text-to-image diffusion model. The input images of size $512$ are $8\times$ downsampled to latents of size $64 \times 64$, which are concatenated with the relative camera raymap and a binary mask that indicates whether or not the image has been observed. A 2D U-Net with temporal connections is utilized for building the latent diffusion model. After each residual block with resolution $\leq 32 \times 32$, we inflate the original 2D self-attention (spatial) of the text-to-image model to be 3D self-attention (spatiotemporal). We remove the text embedding conditioning of the original model.

we sample $128 \times 128$ patches of rays and train with a global batch size of 1M that takes $4$ minutes to train. For single image to 3D, we use $32 \times 32$ patches of rays and a global batch size of 65k that takes $55$ seconds to train. Learning rate is logarithmically decayed from $0.04$ to $10^{-3}$. The weight of the perceptual loss (LPIPS) is set to $0.25$ for single image to 3D and few-view reconstruction on RealState10K, LLFF and DTU datasets, and to $1.0$ for few-view reconstruction on CO3D an MipNeRF-360 dataets.

**Distance based weighting.** We design a weighting schedule that upweights views closer to captured views in the later stage of training to improve details. Specifically, the weighting is given by a Gaussian kernel: $w \propto \exp\left(-bs^2\right)$, where $s$ is the distance to the closest captured view and $b$ is a scaling factor. For few-view reconstruction, $b$ is linearly annealed from $0$ to $15$. We also anneal the weighting of generated views globally to further emphasize the importance of captured views.

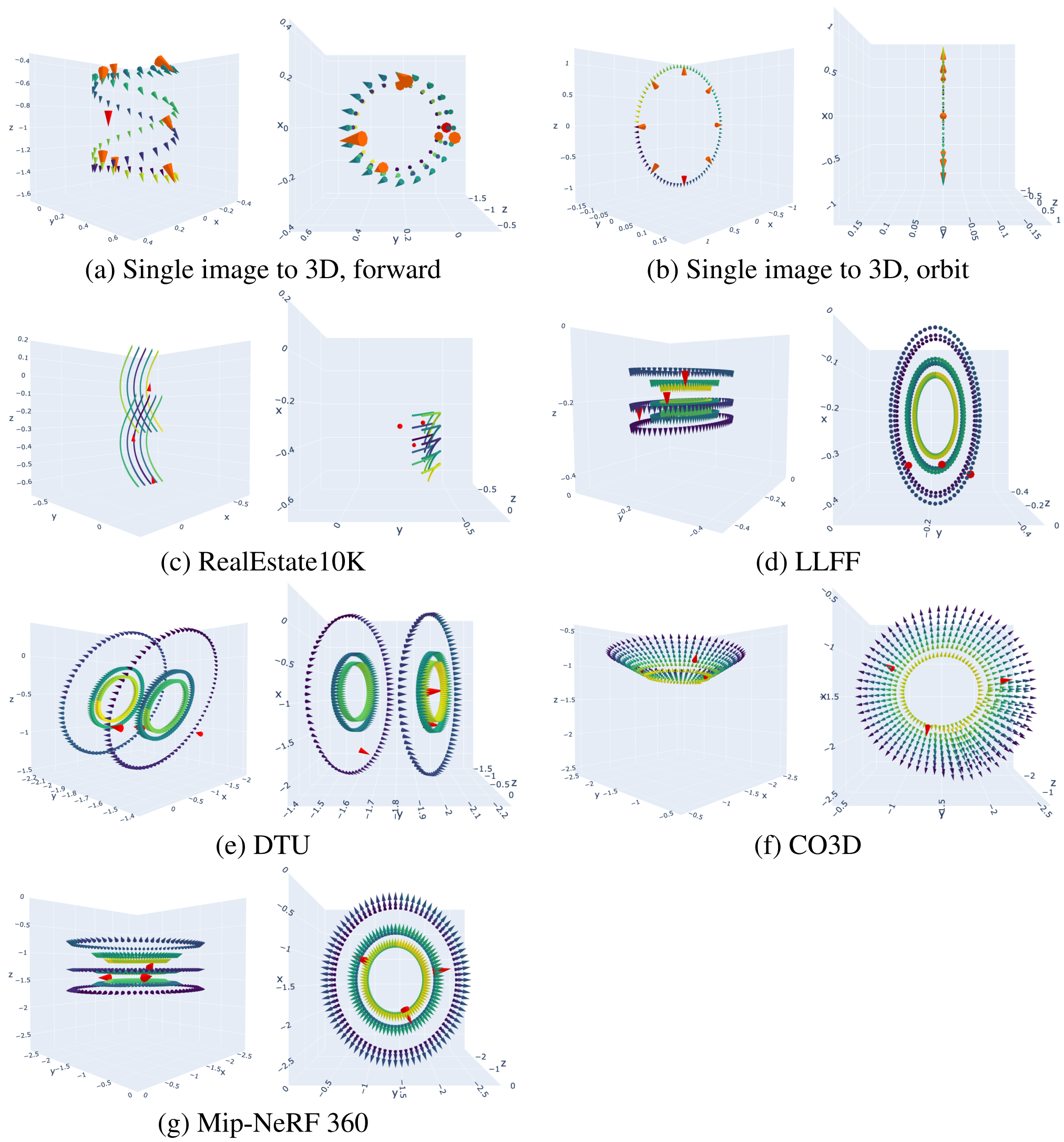

(a) Single image to 3D, forward
(b) Single image to 3D, orbit
(c) RealEstate10K
(d) LLFF
(e) DTU
(f) CO3D
(g) Mip-NeRF 360

Figure 8: **Camera trajectories for generating novel views.** Within each panel, left shows the side view and right shows the top view of the trajectories, colored by indices of views. (a)-(b): two types of trajectories used by single image to 3D. Observed view is highlighted in red, while anchor views are highlighted in orange. (c)-(g): trajectories used by 3D reconstruction. 3 input views are highlighted in red.

