# OpenReview forum: "CAT3D: Create Anything in 3D with Multi-View Diffusion Models"
_NeurIPS.cc/2024/Conference — NeurIPS 2024 oral_

### Official Review · Reviewer_Twzm · 2024-06-30

**Soundness:** 4
**Presentation:** 3
**Contribution:** 4
**Rating:** 7
**Confidence:** 5

**Summary:**

This paper proposed CAT3D, a pipeline that enabled the production of 3D representations from one or a few input views. CAT3D comprises a multi-view generation model to synthesize novel images from different viewpoints and a Zip-Nerf to achieve 3D reconstruction based on generated views. The 3D results shown in this paper are very impressive with high quality, and the authors provided sufficient experiments to verify the effectiveness of CAT3D.

**Strengths:**

1. CAT3D achieves impressive results of 3D generation.
2. The overall pipeline of multi-view generation is straightforward, yet effective with good robustness as verified in the experiments.
3. Although CAT3D is trained with constrained multi-view images (no background: objaverse; object-centric: CO3D, MVImgNet, most indoor scenes: RealEstate10k), it still enjoys good generalization.
4. This paper can be seen as evidence to verify that scaling up 3D generation through multi-view synthesis is feasible.
5. The ablation study is convincing and sufficient.

**Weaknesses:**

1. Most techniques have been proposed by previous works, including the raymap, Nerf with LPIPS loss (IM-3D). But I think this point is not the main issue of this paper, while CAT3D proves the scalability of combining all these techniques together.
2. Since ZIP-Nerf is utilized in CAT3D for dense 3D reconstruction, it is unclear whether the results shown in Figures 1, 2, 4, and 5 in the paper are derived from ZIP-Nerf or generated from the multi-view diffusion process. Are all these results exclusively from ZIP-Nerf? If not, it would be beneficial to provide more results directly from the diffusion model. This distinction is important, especially considering the potential inconsistencies noted in the limitations section.
3. It is not inappropriate to provide the inference efficiency with 16 A100 GPUs (Line573) only, which is not a regular setting for most users. The authors should provide the inference time with one GPU. More importantly, it is unclear whether the efficiency results from Table 2 are all fairly compared with one GPU or the same hardware condition.
4. CAT3D seems only being trained with 1cond+7target and 3cond+5target. Could it address the combination of arbitrary conditions and targets without fine-tuning? For example, 4cond+4target or 2cond+6target.
5. CAT3D enjoys good generalization, which is just trained on objaverse, CO3D, MVImgNet, and RealEstate10k. However, these datasets are all constrained (no background: objaverse; object-centric: CO3D, MVImgNet, most indoor scenes: RealEstate10k). How to confirm the generalization of full-model trainable StableDiffusion, especially for some text-to-image samples shown in the supplementary? This point is not mentioned in the main paper.
6. Unfortunately, the authors did not promise to release the code as shown in the checklist. Therefore, some implementation details should be further clarified as mentioned in the "Questions".

**Questions:**

Some unclear implementation:

1. Lines 190-191 are unclear. How to deal with images with different aspect ratios during training and inference?
2. Missing details about the shift of the noise scheduler.
3. What is meant by "drop the text embedding"? Does it imply using an empty string "" as the input text for all samples, or does it completely remove all cross-attention layers? Additionally, how is stability maintained when all cross-attention layers are removed at the beginning of training?

Some other minor questions:
1. As mentioned in the paper, a mask channel is concated to the inputs to distinguish conditions and targets. Why not try to use SD-inpainting as the initialization to cover this task?
2. Could 3DGS be converged with fewer generated images? Generating 80/720 views is too costly in my opinion.

**Limitations:**

The authors discuss the limitations. However, no qualitative limitations are shown in the paper.

---

> ### Author Rebuttal · Authors · 2024-08-07
>
> Thank you for your time and careful review of our work. Below we address your questions and weaknesses mentioned:
>
> > Most techniques have been proposed.
>
> We agree with you that CAT3D leverages existing techniques for individual components of the system. The innovation of CAT3D lies in the effective integration and scalability of those components: a multi-view diffusion model, efficient sampling strategies, and a robust 3D reconstruction pipeline, resulting in a powerful and practical system for 3D content creation given any number of input views. CAT3D decouples the generative prior from the 3D extraction process, which not only contributes to its efficiency and simplicity, but also allows for potential improvements in either component without affecting the other in future research.
>
> > Results from the 3D reconstruction / multi-view diffusion.
>
> All results in the main text are renderings of optimized mip-NeRF models, but the Appendix also includes quantitative results of images produced by the multi-view diffusion model (Table 3), and the anonymous website in the supplementary contains qualitative samples from the multi-view diffusion model. We will clarify this in the revised draft, and are happy to include more results from the diffusion model in the main text if the reviewer deems this to be useful.
>
> > Run time on one GPU.
>
> Thank you very much for pointing out the differences in timings. We have added a note to the Table indicating that our timings take place on 16xA100 GPUs, and are working to evaluate the timing for a single GPU. One advantage of our approach is that it can benefit from parallelism when generating novel views, while other methods built around distillation and feedforward prediction do not.
>
> > Support arbitrary conditions and targets.
>
> Interestingly, we find our model can work on test settings different from what the model is trained for to some extent. For example, we find that doubling the target views or tripling the conditional views still works (see Figures 2 and 3 in the rebuttal PDF). We still anticipate training or fine-tuning the model on the actual test settings would lead to better results.
>
> > Full-model generalization.
>
> This is a good question and a valid concern. We tried to verify the generalization ability of our model empirically by, e.g., testing our model on a wide variety of captured or generated images that are far out-of-distribution from what the model is trained on (see gallery.html in the supplementary). In Table 3, we observe that the model initialized from the pretrained text-to-image model performs better than the model initialized from scratch quantitatively (suggesting that the pre-trained priors are still present or useful in some capacity). To further maintain the generalization ability, a possible future direction is to jointly train the model on our datasets mixed with the text-to-image data that the model is pretrained on.
>
> > Images with different aspect ratios
>
> While the multi-view latent diffusion model we trained only supports 512x512 images, we found the model still performed well when padding non-square images to square. However, this method often reduces resolution, so we also run our model on a square-cropped version of the inputs, and then compose the square-cropped outputs with the edges from the padded outputs to create a different aspect ratio image. We will add these details to the Appendix.
>
>
> > shift of the noise scheduler.
>
> As we describe in the text (lines 143-145), we add $\log(N)$ to the log signal-to-noise ratio, where $N$ is the number of target images. More concretely, assuming for the base text-to-image model each time step $t$ is mapped to a log signal-to-noise ratio $\log \lambda_t$; then for our model, $t$ is mapped to $\log \lambda_t - \log(N)$. In settings with a mixed number of target views (e.g. 5 and 7 target views), we pick $N$ to be the smallest number of target views (i.e., 5). The logic is similar to [67] where the noise schedule is shifted for higher resolution generation (see Eqn (4) in [67]).
>
> > Drop the text embedding.
>
> It is removed completely for model simplicity. While architecture changes like this can impact the stability of fine-tuning, we found that a learning rate warmup is sufficient to mitigate potential instabilities.
>
> > Mask and SD inpainting.
>
> The mask we used is to specify conditioning vs. target images. It is constant per image (i.e., not spatially varying), whereas the mask in SD inpainting is used for indicating unknown pixels within an image, which may not align with our task.
>
> > 3DGS.
>
> As far as we know, all radiance field models including 3DGS and NeRFs are data hungry and 3DGS doesn’t necessarily require fewer captured (or generated) images. It’s worth noting that generating novel views is not the main computational bottleneck in the whole system. For example, in the single-image-to-3D setting, it takes 5 seconds to generate 80 views while it takes 55 seconds to run 3D scene extraction.

---

> > ### Comment · Reviewer_Twzm · 2024-08-09
> > **Thanks for the rebuttal**
> >
> > Thanks for the rebuttal. Most concerns about implementation details have been addressed; however, I still have some questions that warrant further discussion.
> > 1) It is interesting to see that CAT3D can be generalized to various condition and target views. Is this capability attributable to that no positional encoding is used in CAT3D? To my knowledge, video models with positional encoding fail to be extended to inference with arbitrary lengths.
> > 2) For the shift of the noise scheduler, I think the conclusion of using log(N) to shift the scheduler (N is the number of target images) is not convincing.
> > Because when CAT3D is trained with 5 and 7 target views, it is just evaluated with N=5 in the scheduler. This only confirms that using large N helps multi-view training. The specific relationship between the number of views and the scheduler has not been thoroughly assessed in this paper, leaving the log(N) shifting here somewhat ambiguous.
> > Furthermore, it's important to note that multi-view images should not simply be equated with high-resolution images, given that the overlap among multi-view images can vary greatly and is stochastic.
> > 3) I think that providing multi-view generation from diffusion before Nerf learning is very important to confirm the capacity limit of stablediffusion.
> > Even if the model itself cannot be publicly released, sharing samples—such as  80 views generated from Stable Diffusion before NeRF optimization (in the gallery.html)—would greatly benefit the community. This would enable researchers to reproduce performance based on these multi-view images in their own NeRF optimizations.

---

> ### Author Response · Authors · 2024-08-09
> **Thanks for the reply**
>
> Thanks for the reply. Please find our response below:
>
> > It is interesting to see that CAT3D can be generalized to various condition and target views. Is this capability attributable to that no positional encoding is used in CAT3D? To my knowledge, video models with positional encoding fail to be extended to inference with arbitrary lengths.
>
> While CAT3D does not use an embedding of time (e.g. positional encoding of the time index for each frame), it does use an embedding of camera pose (via the raymap). Unlike time embeddings where during training you only see a small but finite number of time embeddings, we see a much larger continuous set of pose embeddings which may aid in generalization. In video and language models, one still can get generalization depending on how you structure and interpolate the time embeddings (see e.g. [1, 2]).
>
> [1] Chen, Shouyuan, et al. "Extending context window of large language models via positional interpolation."
>
> [2] Kazemnejad, Amirhossein, et al. "The impact of positional encoding on length generalization in transformers."
>
>
> > For the shift of the noise scheduler, I think the conclusion of using log(N) to shift the scheduler (N is the number of target images) is not convincing. Because when CAT3D is trained with 5 and 7 target views, it is just evaluated with N=5 in the scheduler. This only confirms that using large N helps multi-view training. The specific relationship between the number of views and the scheduler has not been thoroughly assessed in this paper, leaving the log(N) shifting here somewhat ambiguous. Furthermore, it's important to note that multi-view images should not simply be equated with high-resolution images, given that the overlap among multi-view images can vary greatly and is stochastic.
>
> It is common practice in training video diffusion models to shift the noise schedule based on the number of target frames, to compensate for the amount of redundant information which may exist across pixels. This is similar to what is done when increasing a models spatial resolution. In reality, the amount of redundant information in a video is a function of the amount of camera and scene motion, i.e., how many pixels are similar or the same across frames, but the number of frames serves as a reasonable approximation. In practice, our model is adapted from a single image diffusion model, and therefore modifying the noise schedule from that base model is necessary, since supervising and predicting multiple frames strictly has more redundant information. And indeed, in practice, we found shifting noise (while training and sampling) improves the quality of results. It is true that we don’t dynamically adjust the schedule based on the number of target frames, we just use the same shift of log(5) for both 5 and 7 target frames, but we found that the difference between that shift in practice is small (e.g. the average LPIPS on the in-domain diffusion samples is 0.235 for shifting log(5) vs. 0.240 for shifting log(7)). The precise formula for log(N), while not explicitly defined in prior work, was something we derived approximately from the numerical noise schedule information provided in the video model literature [3,4]. We will add this detail to the paper. Future work on multi-view models may want to instead condition the model on logSNR and use different shifts when training and sampling with different numbers of frames.
>
> [3] Blattmann, Andreas, et al. "Align your latents: High-resolution video synthesis with latent diffusion models."
>
> [4] Blattmann, Andreas, et al. "Stable video diffusion: Scaling latent video diffusion models to large datasets."
>
>
> > I think that providing multi-view generation from diffusion before Nerf learning is very important to confirm the capacity limit of stablediffusion. Even if the model itself cannot be publicly released, sharing samples—such as 80 views generated from Stable Diffusion before NeRF optimization (in the gallery.html)—would greatly benefit the community. This would enable researchers to reproduce performance based on these multi-view images in their own NeRF optimizations.
>
> Great point, we will include more samples alongside our NeRF results in the project page (and will also need to combine those videos with a serialized form of the camera pose trajectories).

---

> > ### Comment · Reviewer_Twzm · 2024-08-10
> > **Thanks for the reply**
> >
> > Thanks for the reply. My concerns are all addressed. So I raise my score to 7.

---

> > > ### Author Response · Authors · 2024-08-12
> > >
> > > Thank you very much!

---

### Official Review · Reviewer_xucG · 2024-07-11

**Soundness:** 4
**Presentation:** 3
**Contribution:** 4
**Rating:** 8
**Confidence:** 5

**Summary:**

In this paper, the authors propose a two-stage method for 3D creation. Specifically, they introduce a multi-view diffusion model to generate novel views given observed input views. Then using these views, they perform a robust 3D reconstruction using a Zip-NeRF variant. To generate consistent views, they also design a data-dependent sampling strategy. In addition, they also conduct extensive experiments to validate the effectiveness of the proposed method.

**Strengths:**

+ The proposed method is simple but effective. It also obtains a superior 3D creation on scenes.
+ The experiments are exhaustive and validate each components comprehensively, making this paper solid.
+ This paper is easy to follow and provides much details. Thus, it is easy to reproduce it.

**Weaknesses:**

- It is certain that using larger diffusion models can boost the performance. But it is interesting to showcase the improvement trend with increased diffusion models.

**Questions:**

See the Weaknesses.

**Limitations:**

Yes, the authors have adequately addressed the limitations and potential negative societal impact of their work.

---

> ### Author Rebuttal · Authors · 2024-08-07
>
> Thank you for your time and feedback on our work.
>
> > larger diffusion models can boost the performance
>
> We certainly expect that larger models will lead to improved performance and can generate more consistent novel views. One relevant piece of evidence: we experimented with different model variants (Table 3) and found that increasing the amount of computation (e.g., by adding 3D attention operations at every resolution layer of the UNet) can improve performance (albeit at the expense of efficiency in training and sampling). We chose one of the smaller models as our showcase result with the intention of striking a balance of efficiency and performance.

---

> > ### Comment · Reviewer_xucG · 2024-08-10
> >
> > Thanks for the rebuttal. My concerns have been addressed. So I retain my score.

---

> > > ### Author Response · Authors · 2024-08-12
> > >
> > > Thank you very much!

---

### Official Review · Reviewer_d7U9 · 2024-07-11

**Soundness:** 3
**Presentation:** 4
**Contribution:** 3
**Rating:** 7
**Confidence:** 4

**Summary:**

This paper introduces CAT3D, a novel approach for generating 3D representations from a flexible number of input images. The authors tackle the challenge of limited input data, a common bottleneck for 3D reconstruction, by leveraging the power of multi-view diffusion models. Their method generates a collection of novel viewpoints consistent with the input, effectively transforming a sparse-view reconstruction problem into a more manageable dense-view scenario. These generated views are then fed into a robust 3D reconstruction pipeline based on a modified Zip-NeRF to produce the final 3D model.

The authors demonstrate impressive results on various benchmarks, showcasing CAT3D's ability to handle single images, sparse multi-view captures, and even text prompts as input. The method exhibits state-of-the-art performance on few-view 3D reconstruction tasks, outperforming existing methods in terms of both speed and accuracy on established datasets. While single-image 3D generation shows promise, the authors acknowledge the performance is not yet on par with leading methods specifically designed for that task, particularly for single objects. The paper presents a compelling advancement in 3D content creation by unifying different input modalities within a single framework and showcasing significant efficiency gains.

**Strengths:**

- The paper presents a novel approach to 3D content creation by reframing the challenge of sparse input as a view generation problem. This core idea of generating the data needed for robust reconstruction is a valuable contribution to the field.

- The authors demonstrate the effectiveness of CAT3D through comprehensive experiments on established benchmarks. Their results on few-view 3D reconstruction tasks are particularly impressive, showcasing state-of-the-art performance on standard metrics and surpassing existing techniques in terms of both speed and accuracy. The ablation study is well-executed, providing valuable insights into the contribution of different components of their method.

- The paper is generally well-written and easy to follow. The authors clearly motivate their work, provide sufficient background information, and describe their methodology in a structured manner. The figures are informative and complement the textual descriptions well.

- The ability to generate high-quality 3D content from a flexible number of input images, as CAT3D aims to achieve, has substantial practical significance. This flexibility is highly desirable for various applications. The demonstrated speed improvements over existing iterative optimization methods further add to its potential impact by enabling more efficient workflows.

**Weaknesses:**

- While CAT3D aims to handle sparse inputs, its dependence on calibrated camera poses, presents a significant limitation. How does performance degrade with increasing sparsity and decreasing pose accuracy?

- The paper relies on manually designed camera trajectories for novel view synthesis, which limits practicality and scalability. The authors briefly mention adapting trajectories based on scene characteristics but provide no concrete details. Developing an automated trajectory selection or optimization procedure, potentially guided by learned priors or scene understanding techniques, would significantly enhance the method's value and broader applicability.

- The paper, at times, overstates its accomplishments (e.g., "achieving state-of-the-art performance across nearly all settings") and does not adequately address its limitations. A more nuanced and critical self-assessment would strengthen the work.

**Questions:**

- Could the authors provide a more quantitative assessment of how performance degrades with increasing pose noise or sparsity?

- Have the authors considered incorporating an automatic trajectory optimization scheme within CAT3D?

- While open-sourcing the code and trained models is ideal, could the authors at least elaborate on their plans for sharing their work and facilitating reproducibility? Providing more details about the training procedure and hyperparameters would also be beneficial.

**Limitations:**

The authors list several limitations, including the reliance on constant camera intrinsics during training, the expressiveness limits of the base text-to-image model, the small number of output views, and the need for manual camera trajectories. However, the discussion lacks concrete examples or quantifiable measures of the limitations.

---

> ### Author Rebuttal · Authors · 2024-08-07
>
> Thank you for the careful reading and kind words. Below we address your questions and weaknesses mentioned:
>
> > performance with increasing sparsity and decreasing pose accuracy
>
> In terms of performance while varying the number of input views with accurate camera poses, Table 1 includes qualitative results for 3, 6, and 9 view sparse reconstruction. The CAT3D model was not trained with missing or noisy poses, but some of our training data (CO3D) has imperfect pose which leads to some small amounts of robustness. To verify this, we conducted an experiment by perturbing the camera rotations of the input conditioning 3 views by a certain number of degrees, and measuring the error between the generated and ground truth target images. See Figure 1 in the rebuttal PDF for average PSNR across varying amounts of rotation perturbation on the MipNerf-360 dataset. Our model can handle small rotation perturbation. Fine-tuning the model with more perturbations on camera pose should lead to improved performance in this setting, which would be an interesting direction for future work.
>
> > manually designed camera trajectories
>
> We agree with the reviewer that jointly learning a trajectory model or inferring trajectories given scene content is an exciting direction for future work. Empirically, we found a handful of simple heuristic trajectories worked well for our experiments, but ideally the trajectories should cover the scene without being placed inside of objects or walls as mentioned in line 167-169.
>
> > overstates its accomplishments
>
> We are happy to revise language to better reflect limitations. The “state-of-the-art performance across nearly all settings” is referring to Table 1, where our method indeed exhibits stronger performance than all prior work. Were there any other passages of text that you would like us to alter? We discuss several limitations in the discussion section, and can expand for the camera ready (especially discussing the challenges we mentioned above regarding trajectory selection).
>
> > reproducibility
>
> In the Appendix, we aimed to provide all details necessary to reproduce CAT3D on top of an open-source latent diffusion model. If there are any additional details that the reviewer thinks are missing, we are happy to include them in an updated draft.
>
> > constant camera intrinsics, other limitations
>
> It’s worth noting that our model does not rely on constant camera intrinsics during training. This is an artifact of our current training dataset where the camera intrinsics are approximately fixed for each scene. If we were to capture and include additional training data with camera intrinsics varying within a scene, we expect our model would be able to perform camera intrinsic manipulation. As far as limitations in expressiveness of the base model, one notable example is that our model performs poorly on human faces, as the base model was not trained on much human data. A showcase of the limitation of producing a small number of output views can be seen in the Supplementary website, where the generated spin video is clearly not perfectly 3D consistent. The need for manual camera trajectories is shown in Fig. 8, by the fact that it was necessary to create different types of trajectories based on characteristics of different datasets. We will further emphasize these points in the paper text.

---

> > ### Comment · Reviewer_d7U9 · 2024-08-13
> > **Thanks for the rebuttal**
> >
> > Thanks for the clarification. After reading the rebuttal and other reviews, I decide to retain my score (accept). I believe the work deserves to be presented at NeurIPS.

---

### Official Review · Reviewer_1Rgc · 2024-07-13

**Soundness:** 3
**Presentation:** 4
**Contribution:** 3
**Rating:** 6
**Confidence:** 5

**Summary:**

The objective of this paper is to achieve single-view or few-view to 3D. The core of their method lies in a multi-image-based diffusion model that leverages 3D attention and raymap encoding for the camera poses.  This setup is different from concurrent work, IM-3D, which repurposes video generation model to achieve 3D, or ReconFusion, which iteratively refine novel view with diffusion model conditioned on PixelNeRF embeddings. Experiments are conducted against several competitive Baselines, like IM-3D and ReconFusion. Result-wise, their proposed method showcases a nice balance between quality and efficiency.

**Strengths:**

1. The quality of the generated/reconstructed 3D scene is state-of-the-art, and it is more efficient compared with ReconFusion and other iterative methods.

2. The proposed multi-view diffusion model is effective. In LRM related literature, image-based diffusion is tricky to generate multi-view consistent. The proposed CAT3D certainly gives better multi-view image generation quality without explicit maintaining a 3D representation, which can benefit many related tasks.

3. The paper is clearly motivated, and the experiments are carefully designed and reported.

**Weaknesses:**

1. It seems cumbersome to generate a large number of viewpoints(Line:174)  by doing anchor first, and enriching frames in between by repeated running CAT3D. Indeed, more view generated together from the multi-view diffusion is good. But running it iteratively will still produce inconsistencies over runs over different set of camera viewpoints, as is can be seen from Fig.6.

2. The trajectory shape seems to influence the quality of the multi-view diffusion. This limits its generalizability to arbitrary/scattered image viewpoints towards the same scene target, which are common in daily life.

**Questions:**

1. What happens if we only use anchor views to run recon in step2? How the quality compared with the current setup?

2. What if we densify view through some existing video interpolation model (instead of re-run CAT3D)? Would be more efficient and better quality?

**Limitations:**

Yes.

---

> ### Author Rebuttal · Authors · 2024-08-07
>
> Thank you for your time and careful review of our work. Below we address your questions and weaknesses mentioned:
>
>
> > cumbersome to generate a large number of viewpoints
>
> We agree that jointly generating all target frames from the multi-view diffusion model would enable more consistent samples. However, simultaneously generating a large number of frames with video diffusion-like architectures is still an active area of research, with many SOTA text-to-video models using autoregressive generation in time to produce a larger number of frames (while losing consistency due to limited context length). By splitting the generation into anchors and then independent blocks of generation at different camera locations, we can efficiently generate a large number of frames *in parallel*. Building efficient architectures that can support the joint generation of a larger number of frames is an exciting direction for future research.
>
> > trajectory influences quality
>
> Our multi-view model can generalize to arbitrary camera trajectories, and we show results for other input capture types, e.g., forward-facing scenes like LLFF which contain several images pointed towards the same scene target. It is true that the model performance varies for different camera trajectories, and this is likely due to some of the biases inherited from our relatively limited training dataset. Mixing in training data with more diverse camera distributions could be an interesting future direction.
>
> > only use anchor views for reconstruction
>
> Our model was only trained to produce at most 8 views total (conditioning + target), and the 3D reconstruction methods we use (i.e., Zip-NeRF) are seldom able to reconstruct plausible geometry from 8 views (as a reference, see results with Zip-NeRF and 9 *real* views which are strictly better than 8 generated views in Table 1). Including additional frames through AR generation is critical to yield the high-quality reconstruction and NVS results we present in the paper (see also Fig. 6 that compares 80 vs. 720 frames).
>
> > densify views through video interpolation
>
>
> Using a video interpolation model to generate frames is an interesting idea. However, it is not clear how to use such a model to incorporate more than 2 frames, and how to leverage the resulting frames in 3D reconstruction without explicit camera control. It’s also not clear whether this approach would be more efficient or would yield better quality results: if using an architecture similar to ours, the compute cost would be similar, and one would also have to estimate camera poses for the resulting frames (which adds an additional computational expense that our pose-conditioned multi-view diffusion models do not incur).

---

> > ### Comment · Reviewer_1Rgc · 2024-08-13
> >
> > Thanks for the detailed clarification! It is one solid work. I retained my score

---

### Author Rebuttal · Authors · 2024-08-07

We thank all the reviewers for their time and critical feedback to improve our work. We appreciate that the reviewers found our method simple, effective, and efficient, leading to a “compelling advancement in 3D content creation.” We address individual questions below, but first highlight some shared concerns.

One of the biggest limitations of our work is the need to specify a desired camera trajectory, along which to generate novel views that supervise 3D reconstruction. Choosing the correct trajectory can be challenging for complex scenes, and we believe this is an exciting direction for future work. We found that simple orbit trajectories and forward-facing explorative trajectories can be useful across a wide variety of inputs (see Supplementary website), but for more complicated scenes a more automated strategy for bespoke trajectory selection would be useful.

Regarding our autoregressive sampling strategy, we found that first producing anchors and then generating sets of views in parallel produced mostly consistent views while drastically accelerating sampling time. This strategy is distinct from video diffusion models where all frames are generated at once, and allows us to produce 3D scenes faster.

We’ve aimed to ensure all details needed to reproduce this work are included in the text, and we are happy to iterate with reviewers to ensure this is the case.

---

### Decision · Program_Chairs · 2024-09-25

**Decision:**

Accept (oral)

**Comment:**

This paper was reviewed by four experts in the field.  Based on the reviewers' feedback, the decision is to recommend the paper for acceptance.  The reviewers did raise some valuable comments that should be addressed in the final camera-ready version of the paper. The authors are encouraged to make the necessary changes to the best of their ability.    We congratulate the authors on the acceptance of their paper!